# INFERRING FLUID DYNAMICS VIA INVERSE RENDERING

## ABSTRACT

Humans have a strong intuitive understanding of physical processes such as fluid falling by just a glimpse of such a scene picture, *i.e.*, quickly derived from our immersive visual experiences in memory. This work achieves such a photo-to-fluid-dynamics reconstruction functionality learned from unannotated videos, without any supervision of ground-truth fluid dynamics. In a nutshell, a differentiable Euler simulator modeled with a ConvNet-based pressure projection solver, is integrated with a volumetric renderer, supporting end-to-end/coherent differentiable dynamic simulation and rendering. By endowing each sampled point with a fluid volume value, we derive a NeRF-like differentiable renderer dedicated from fluid data; and thanks to this volume-augmented representation, fluid dynamics could be inversely inferred from the error signal between the rendered result and ground-truth video frame (*i.e.*, inverse rendering). Experiments on our generated Fluid Fall datasets and DPI Dam Break dataset are conducted to demonstrate both effectiveness and generalization ability of our method.

## 1 INTRODUCTION

Simulating and rendering complex physics (*e.g.*, fluid dynamics, cloth dynamics, hair dynamics) are major topics in computer graphics, with prevailing applications in games, movies and animation industry. To pursue better adaptability, generalization ability and efficiency, many works (Wiewel et al., 2019; Kim et al., 2019; Li et al., 2019; Ummenhofer et al., 2019; Sanchez-Gonzalez et al., 2020; Pfaff et al., 2021; Wandel et al., 2022; de Avila Belbute-Peres et al., 2018; Hu et al., 2019; Loper & Black, 2014; Liu et al., 2019; Mildenhall et al., 2020; Liu et al., 2020) pay attention to either learning-based (differentiable) simulation or rendering recently, especially to learnable simulation. However, most of these works need large amounts of ground-truth data simulated using traditional physical engines to train a robust and general learning-based simulator.

Meanwhile, most existing works rely on the traditional pipeline; namely, it requires stand-alone modules including physical dynamics simulation, 3D modeling/meshization and rasterization/ray-tracing based rendering, thus the complication makes it tedious and difficult for live streaming deployment. Few works (Wu et al., 2017a; Guan et al., 2022; Li et al., 2021b) attempt to associate simulation and rendering into an integrated module. VDA (Wu et al., 2017a) proposes a combined framework, where a physics engine and a non-differentiable graphic rendering engine are involved to understand physical scenes without human annotation. However, only rigid body dynamics are supported. NeuroFluid (Guan et al., 2022) is a concurrent work that proposes a fully-differentiable two-stage network and grounds Lagrangian fluid flows using image supervision. However, image sequences provide supervision that only reveals the changes in fluid geometry and cannot provide particle correspondence between consecutive frames (*i.e.*, temporal/motion ambiguities) to support Lagrangian simulation. Moreover, lots of spatial ambiguities between particle positions and spatial distribution of fluid (*e.g.*, different particles positions may result in the same fluid distribution used for rendering) are introduced by the proposed Particle-Driven NeRF (Guan et al., 2022). In reverse rendering, ambiguity is a terrible problem that makes the optimization process prohibitively ill-posed (Zhao et al., 2021). Thereby, NeuroFluid can only overfit one sequence every time and cannot learn fluid dynamics. (Li et al., 2021b) develops a similar framework, while it learns dynamics implicitly on latent space and represents the whole 3D environment using only a single vector. Such a rough and implicit representation limits its ability for generalization. **We make detailed discussions and comparisons with (Guan et al., 2022; Li et al., 2021b) in Section 4.4.**

To explicitly address these fundamental limitations, in this work, we propose an end-to-end/coherent differentiable framework integrating simulation, reconstruction (3D modeling) and rendering, with image/video data as supervision, aiming at learning fluid dynamic models from videos. In a nutshell, a fluid volume augmented representation is adopted to facilitate reconstruction-dynamics linkage between simulation and rendering. More concretely, we design a neural simulator in the Eulerian view, where 3D girds are constructed to save the velocity and volume of fluid. In addition to advection and external forces application, we develop a ConvNet-based pressure projection solver for differentiable velocity field updating. Naturally, the fluid volume field is updated using advection based on the updated velocity field. Note that the advection is an efficient and differentiable operator that involves back-tracing and interpolation. In the meantime, as the Euler 3D grid saves the volume of fluid, a NeRF-like neural renderer is thus proposed to capture the geometry/shape information of fluid which retrieves supervision signals from images taken from multiple views. Specifically, we assign a fluid volume value to each point sampled from emitted rays by performing trilinear interpolation on the fluid volume field. Then the sampled points equipped with fluid volume properties are sent into the NeRF model to render an image. The fluid volume value of each point provides information (about how much fluid material there is) for the renderer to capture the effects of fluid on its density and radiance. The simple and efficient fluid volume representation not only suits a neural Euler fluid engine for fluid motion field estimation but also supports a differentiable NeRF-like renderer relating image with fluid volume, achieving end-to-end error signal propagation to all rendering, fluid volume reconstruction and simulation modules. Note that the fluid field (Euler) representation naturally provides spatial distribution of fluid and better correspondence between consecutive frames, which greatly reduces ambiguities for inverse optimization. The whole forward simulation-modeling-rendering process and the inverse procedure are shown in Figure 1.

Unlike methods (Niemeyer et al., 2019; Pumarola et al., 2021; Tretschk et al., 2021; Ost et al., 2021; Li et al., 2021a; Guan et al., 2022) that typically fit one sequence, our model can be trained on abundant sequences simultaneously and generalizes to unseen sequences with different initial conditions (*i.e.*, initial shape, position, velocity, *etc.*). Besides, we model and simulate fluid in an explicit and interpretable way, which is different from method (Li et al., 2021b) that learns 3D physical scenes in latent space. As shown in Figure 10, such an explicit representation way endows our method with a strong ability to render images in extreme views that are far away from our training distribution and perform scene editing efficiently. We conduct experiments on a part of DPI DamBreak dataset (Li et al., 2019) and two datasets that we generate using Mantaflow (Pfaff & Thuerey) and Blender. Various experiments that involve baseline comparison, representation comparison, future prediction, novel view synthesis and scene editing are performed to prove both effectiveness and generalization ability of our method. Detailed ablation studies are conducted to analyze important components and parameters. Upon acceptance, all code and data will be publicly available. We also discuss the limitations of our work in Appendix A.10.

## 2 RELATED WORK

**Fluid Simulation**. Fluid simulation is a long-standing research area of great interest in science and engineering disciplines. Various classical algorithms (Chorin, 1968; Stam, 1999; Macklin et al., 2014; Fedkiw et al., 2001; Monaghan, 1994; Solenthaler & Pajarola, 2009; Macklin & Müller, 2013; Bender & Koschier, 2015; Bardenhagen et al., 2000; Zehnder et al., 2018; Ando et al., 2015; Zhang & Bridson, 2014; Brackbill et al., 1988; Jiang et al., 2015; Hu et al., 2018) are proposed to facilitate accurate and fast simulation. To pursue better adaptability, generalization ability and efficiency, learning-based fluid simulation (Li et al., 2019; Ummenhofer et al., 2019; Sanchez-Gonzalez et al., 2020; Pfaff et al., 2021; Tompson et al., 2017; Wiewel et al., 2019; Zhu et al., 2019; Kim et al., 2019; Thuerey et al., 2020; Wandel et al., 2020) has attracted increasing attention in recent years. Most of these works usually bypass solving large-scale partial differential equations (*i.e.*, PDE) via efficient convolution operators. Lagrangian flows (Li et al., 2019; Ummenhofer et al., 2019; Sanchez-Gonzalez et al., 2020) usually model the fluid and rigid body as a set of particles with different material types. Graph neural networks (Sanchez-Gonzalez et al., 2020; Pfaff et al., 2021) are also suitable to solve such problems. Euler flows (Tompson et al., 2017; Wiewel et al., 2019; Kim et al., 2019) divide the space into regular grids and save physical quantities (*e.g.*, density, mass, volume, velocity) of material in the divided grids. (Tompson et al., 2017) accelerates Euler fluid simulation using a convolution network to solve pressure projection. (Wiewel et al., 2019) encodes the pressure field into a latent space and designs a LSTM-based network to predict the latent code

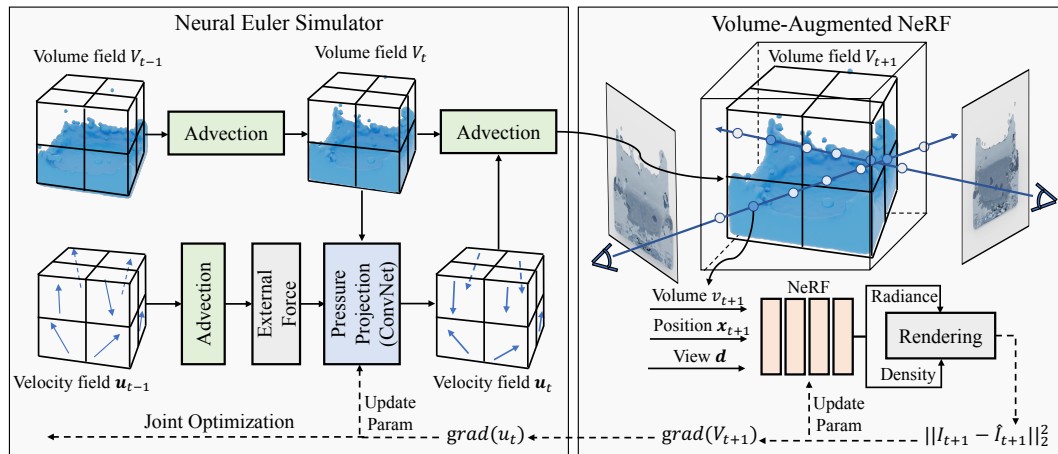

Figure 1: **Framework of our method**. A differentiable neural Euler simulator is constructed to roll out velocity field and fluid volume field. The volume-augmented NeRF serves as a differentiable renderer to synthesize fluid image with the fluid volume as input. These two modules are associated as a fully-differentiable framework and learned with image/video data as supervision. Detailed network structures of the simulator and renderer are shown in Appendix A.1.

for future time steps. Ground-truth pressure fields are usually needed to train such a network. We also propose a differentiable Euler simulator that solves pressure projection using a neural network in this work. However, we associate our simulator with a differentiable renderer and drive the Euler simulator with 2D videos by solving an inverse rendering problem.

**Differentiable Rendering**. Recently, many works (Kato et al., 2018; Insafutdinov & Dosovitskiy, 2018; Loper & Black, 2014; Liu et al., 2019; Li et al., 2018) attempt to turn rendering into a differentiable process. (Kato et al., 2018; Liu et al., 2019) design soft rasterizers to achieve differentiable render. (Li et al., 2018) develops a differentiable monte carlo ray tracer. Neural radiance field (*i.e.*, NeRF) (Mildenhall et al., 2020) is one of the most popular differentiable renderers, where a continuous radiance field is saved in a MLP-based network. To render one image, we just need to sample point on rays and query its volume density and color from the trained NeRF model. Finally, a differentiable volumetric rendering is performed. Novel views can be generated by emitting rays from new positions with new poses. An important advantage of differentiable rendering is that it supports inverse rendering well, *i.e.*, inferring 3D geometry, lighting or material based on the rendered images. In this work, we design a fluid volume augmented NeRF model and take advantage of its ability of inverse rendering to learn fluid dynamics and drive a generalizable simulator.

**Inverse Graphics**. There are many works (Yuille & Kersten, 2006; Zhu & Mumford, 2007; Bai et al., 2012; Kulkarni et al., 2015; Jimenez Rezende et al., 2016; Eslami et al., 2016; Wu et al., 2017b; Yao et al., 2018; Azinovic et al., 2019; Munkberg et al., 2022) that focus on inverse graphics. Recently, some researches (Gupta et al., 2010; Battaglia et al., 2013; Jia et al., 2014; Shao et al., 2014; Zheng et al., 2015; Lerer et al., 2016; Mottaghi et al., 2016; Fragkiadaki et al., 2016; Battaglia et al., 2016; Agrawal et al., 2016; Pinto et al., 2016; Finn et al., 2016; Ehrhardt et al., 2017; Chang et al., 2017; Zhu et al., 2018; Raissi et al., 2019; Li et al., 2020; Chu et al., 2022) on this field pay more attention to reasoning about physical properties of the scene from image/video. In contrast to these methods that infer static properties, we infer a generalizable physical dynamic model via inverse rendering. (Wu et al., 2017a; Guan et al., 2022; Li et al., 2021b) are most relevant works to us. VDA (Wu et al., 2017a) develops a paradigm for understanding physical scenes without human annotations. A physical engine (differentiable or trained with REINFORCE (Williams, 1992)) for scene reasoning and a graphic engine for rendering are constructed to reconstruct the visual data. However, only rigid body dynamics are considered. NeuroFluid (Guan et al., 2022) develops a framework that supports grounding fluid particles from images. Lots of ambiguities are introduced by modeling the fluid in Lagrangian view and make the optimization process very ill-posed. Thus, their model only overfits one sequence every time and cannot learn fluid dynamics. Li et al. (2021b) develops a similar framework that learns dynamic model on latent space. They represent the dynamic 3D environment using only a single vector captured from images, which limits the model to

generalize to cases never seen before. In this work, we propose to reason about generalizable fluid dynamics in an explicit and interpretable way with videos as supervision. Strong generalization ability to unseen initial conditions are demonstrated.

## 3    METHODOLOGY

We develop an end-to-end differentiable framework integrating fluid dynamics simulation, 3D modeling and video frame rendering, with image/video data as supervision. To re-formulate *conventionally* independent simulation and rendering modules into an inter-related fully differentiable pipeline, a fluid volume field augmented representation is adopted, which not only provides information about the interface between air and fluid for better free-surface simulation, but also supports differentiable NeRF-like rendering based on its direct linkage with the density color in neural radiance. Based on this construction, a neural fluid simulator in Eulerian view together with a neural renderer are jointly established, which propagates the error signal between ground-truth video frame and the rendered image all the way from renderer to simulator, thus guiding the fluid to move forward step-by-step. Details about the proposed simulator, fluid representation and renderer, as well as the overall learning framework are introduced as follows and the whole pipeline is shown in the Figure 1.

### 3.1    NEURAL EULER FLUID SIMULATION

Traditional fluid simulation often involves solving incompressible Navier-Stokes equations, one variant of which is formulated as:

$$\frac{\mathrm{D}\mathbf{u}}{\mathrm{D}t} = -\frac{1}{\rho}\nabla p + \nu\nabla^2\mathbf{u} + \mathbf{g}, \tag{1}$$

$$\nabla \cdot \mathbf{u} = 0, \tag{2}$$

where $\frac{\mathrm{D}\mathbf{u}}{\mathrm{D}t}$ is the material derivative of velocity $\mathbf{u}$, $\rho$ is fluid density, $p$ is the pressure (a scale field), $\nu$ is kinematic viscosity and $\mathbf{g}$ is the external force applied to fluid such as gravity. The material derivative of $\mathbf{u}$ represents the rate of change of velocity (*i.e.*, the acceleration), which comes from three components: 1) the derivative of pressure $p$, 2) viscous force (is usually dropped except for highly viscous fluid), 3) external force (*e.g.*, gravity). In the Eulerian view, the space (fluid container) is divided into grid. The physical quantities (*e.g.*, volume $V$, velocity $\mathbf{u}$) of the fluid are saved in the fixed grid. It is free for computing spatial derivatives (finite difference). To solve such a complex dynamic equation, we often split it into three operators, *i.e.*, advection, applying external force and pressure projection (keep the velocity field divergence-free). The challenge lies in solving the pressure projection, which is a large-scale linear system. In this work, we develop a neural pressure projection solver. Specifically, we first perform advection to the fluid volume field $V_{t-1}$ and velocity field $\mathbf{u}_{t-1}$ given $\mathbf{u}_{t-1}$ at time step $t-1$. The advection can be seen as an operator that moves fluid with a constant velocity. In the Lagrangian view, we achieve advection by just moving particles according to its velocity. The physical quantities such as mass and velocity are bound to particles and moved with these particles. However, the grids are fixed in the Eulerian view. To achieve advection and update the physical quantities (volume and velocity) of each fixed grid, we take a commonly used semi-Lagrangian method to do advection, which involves two sub-steps (*i.e.*, backtracing and interpolation). Detailed explanation of advection module is shown in Appendix A.3. We also use Maccormack (MacCormack, 2003) method to perform advection for reducing numerical viscosity. This process is denoted as:

$$V_t = advect(\mathbf{u}_{t-1}, \Delta t, V_{t-1}), \tag{3}$$

$$\mathbf{u}^* = advect(\mathbf{u}_{t-1}, \Delta t, \mathbf{u}_{t-1}), \tag{4}$$

where $\Delta t$ is the time step for simulation. Then we apply gravity to the fluid, *i.e.*, accelerating the velocity field using gravitational acceleration $\mathbf{g}$:

$$\mathbf{u}^{**} = \mathbf{u}^* + \mathbf{g}\Delta t. \tag{5}$$

Then the updated velocity field $\mathbf{u}^{**}$ are sent into a ConvNet to regress the velocity difference introduced by the pressure that comes from the interaction between fluid elements, fluid and rigid bodies, fluid and other medias, *etc.*

To better handle scenes of free surface fluid simulation (*i.e.*, the Euler space contains both fluid and air), for each grid in the Eulerian view, we augment a volume value to indicate its material

property, *i.e.*, a real value in the range of 0 to 1.0 with 0 representing it contains no fluid and 1.0 representing full of fluid, and the whole space (container) could be regarded as a volume field. The advantage of using this augmented representation is obvious. First, this volume field illustrates the fluid volume distribution in the space explicitly, hence it tells the neural rendering network where the interface between fluid and air is to facilitate high fidelity material interface rendering. Second, in the Euler framework, the information about the material interface is also helpful in solving the pressure projection step. We, therefore, take the volume of the fluid $V_t$ as the additional input of the ConvNet for pressure projection. In addition, we assign each grid an input feature vector that represents the distance from it to the container boundary. Given all these inputs, the ConvNet regresses the $\Delta \mathbf{u}$ induced by the pressure: $F_\Theta : (\mathbf{u}^{**}, V_t) \to (\Delta \mathbf{u})$. The updated velocity at next step is written as:

$$\mathbf{u}_t = \mathbf{u}^{**} + \Delta \mathbf{u}. \tag{6}$$

Moreover, our fluid should be incompressible and the pressure projection has to keep the velocity divergence-free. To this end, we apply regularization to the updated velocity field $\mathbf{u}_t$, *i.e.*, to add a penalty for its divergence when training the pressure projection network. This design helps the simulator to do better in alleviating volume loss. The framework of our Euler fluid simulation is presented in Figure 1. The detailed structure of the ConvNet is presented in Appendix.

## 3.2 VOLUME-AUGMENTED RADIANCE FIELD

Note that the fluid volume representation also serves as a medium to bridge the simulator and renderer in a joint differentiable way. Based on the volume field, a fluid volume augmented Neural Radiance Field (NeRF) is developed in this work. The original NeRF (Mildenhall et al., 2020) model works only on static scene. Rays represented as $\mathbf{r}(k) = \mathbf{o} + k\mathbf{d}$ are emitted from the cameras to the scene. $\mathbf{o}$ and $\mathbf{d}$ denote the position of camera and view direction respectively. 3D points $\mathbf{x}_i$ ($i = 1, 2, ..., N$) are sampled along each ray with near and far bounds. These sampled points are combined with viewing direction $\mathbf{d}$ and sent into the NeRF model (*i.e.*, MLP-based network) to regress their density $\sigma$ and view-depended colors $\mathbf{c}$. The RGB values of each pixel $C(\mathbf{r})$ are integrated by all sampled points along the corresponding ray using volumetric rendering algorithm. This process is formulated as:

$$\hat{C}(\mathbf{r}) = \sum_{i=1}^{N} T_i \left(1 - \exp\left(-\sigma_i \delta_i\right)\right) \mathbf{c}_i, \tag{7}$$

$$T_i = \exp\left(-\sum_{j=1}^{i-1} \sigma_j \delta_j\right), \tag{8}$$

where $T_i$ denotes the accumulated transmittance along the ray from near bound to point $\mathbf{x}_i$ and $\delta_i$ represents the distance between adjacent samples.

However, the density and the color of each sample point change with the flow of the fluid in our dynamic scenes. This change occurs as the spatial volume distribution of the fluid changes temporally. Hence we propose to bind value of fluid volume for each input sample point for neural radiance rendering, and the volume value provides the rendering network additional information on how much fluid material is contained here. This information is highly correlated with density and radiance of sampled point. Specifically, we assign the volume field of fluid $V$ inferred by the neural Euler simulator into the scene space and perform trilinear interpolation for each sampled point $\mathbf{x}_i$ on rays to capture its corresponding fluid volume value $v_i$. We combine this captured volume value with position of the sample point to learn its volume density and color for rendering image $\hat{I}$, which is written as $F_\Psi : (\mathbf{x}_i, \mathbf{d}, v_i) \to (\mathbf{c}_i, \sigma_i)$. The Euler simulator rolls out the volume of fluid at each time step and guides the NeRF to synthesize a sequence of images and finally renders a fluid video.

## 3.3 RADIANCE-DRIVEN FLUID SIMULATION

Most learning-based fluid simulators are 3D data driven, where large-scale data is generated using traditional computational fluid dynamic (CFD) tools such as SPH, FLIP, *etc.*The simulated particle velocity/position or velocity/pressure field are provided for model training. Based on our fluid volume representation, we naturally associate the differentiable simulator and renderer, further driving

the fluid simulation by scene radiance field learned from images. We have stated that our volume-augmented radiance field network is able to generate corresponding fluid images based on the input fluid volume, *i.e.*, forward computation. When providing a sequence of fluid flow video frames, the volume-augmented NeRF model can enforce the input volume of fluid to be close to real distribution for achieving better rendering results, *i.e.*, to minimize the appearance error between the rendered scene and the ground-truth video frame. Almost no ambiguity is introduced by bridging the simulation and the render using this volume representation. Hence, the error signal between rendered image and ground-truth image provides valid gradients for correcting both the renderer and simulator, through jointly training the neural Euler simulator and volume-augmented radiance field (as the whole pipeline is fully-differentiable). Specifically, we advect the volume field $V_t$ using the velocity field $\mathbf{u}_t$ inferred by the neural Euler simulator and get the updated volume field $V_{t+1} = advect(\mathbf{u}_t, \Delta t, \mathbf{u}_t)$. Then the volume field of fluid $V_{t+1}$ is assigned into the scene space to render image $\hat{I}_{t+1}$ that is supervised by ground-truth image $I_{t+1}$. Namely, the simulator updates its parameters to better solve the pressure projection problem that is formulated as a regression task in our work, hence providing right velocity to advect the fluid volume for better rendering. As a result, a fluid simulator can be learned from videos without other supervision.

### 3.4 OPTIMIZATION DETAILS

The volume-augmented renderer does not have any prior knowledge of the fluid geometry, thus it is hard to directly joint train the simulator and renderer. To endow the volume-augmented NeRF with the ability to capture geometry information from fluid and do inverse rendering preliminarily, we first train the radiance field network with initial volume field of all training sequences (volume of first frames serve as initial conditions for starting fluid simulation). After pre-training, the render model can reason about the geometry information of fluid from ground-truth image and perform rendering preliminarily. Then we jointly train the render model and the simulator on sequence data to learn fluid dynamics, *i.e.*, train the simulator and renderer step-by-step. We summarize the optimization function $\mathcal{L}$ for joint training as follows:

$$\mathcal{L} = \sum_{\mathbf{r}} \|\hat{C}(\mathbf{r}) - C(\mathbf{r})\|_2^2 + \beta \|\nabla \cdot \mathbf{u}\|_2^2, \tag{9}$$

where $\beta$ is set to 1e-6 to control the strength of the regularization term.

## 4 EVALUATION

We conduct experiments on three datasets that involve two Fluid Fall datasets that we generated and DPI Dam Break dataset (Li et al., 2019). Results of simulation and rendering on three datasets are then reported to demonstrate the effectiveness, generalization ability and robustness of our method. Ablation studies are performed to analyze components and hyper-parameters. Experimental settings and more analyses are shown in Appendix A.

### 4.1 DATASETS

**Fluid Fall**. We use synthetic data simulated using a CFD tool Mantaflow (Pfaff & Thuerey) and rendered using Blender. **Two views** for each video are generated. The simulation environment is where fluid objects fall in a cubic container. We vary the initial shape, position and velocity. We generate two versions of datasets in that each sample contains one/two fluid objects randomly sampled from 5 different shapes. Each one of the two datasets contains 100 sequences with 50 steps. 90 sequences are used for training and the rest for test. Please refer to Appendix for more details.
**Dam Break**. We randomly sample 180 sequences with 50 time steps for training and 20 sequences for test from training set and test set of DPI Dam Break dataset respectively. The mesh of each time step is constructed from particles using Marching Cube algorithm and rendered using Blender. **Two views** for each video are generated. Please refer to Appendix for more details.

### 4.2 THE RESULTS OF SIMULATION

In this section, we evaluate our method on simulation. We report results of our method on both training and test set. The mean squared error (MSE) between the predicted volume field and ground-

| Eval set | Method | Fluid Fall (1 obj) | | Fluid Fall (2 objs) | | DPI Dam Break | |
|---|---|---|---|---|---|---|---|
| | | 1-50 steps | 51-60 steps | 1-50 steps | 51-60 steps | 1-50 steps | 51-60 steps |
| Train set | Ours | 0.0289 | 0.0378 | 0.0402 | 0.0533 | 0.0321 | 0.0538 |
| Test set | Ours | 0.0297 | 0.0362 | 0.0427 | 0.0589 | 0.0367 | 0.0622 |
| | Euler-GT | 0.0146 | 0.0206 | 0.0282 | 0.0391 | 0.0216 | 0.0412 |

Table 1: **Results of our method on simulation**. Mean square error (MSE) of predicted fluid volume on all three datasets is presented in the table. We evaluate the simulator on both training set and test set. Results of the simulator trained with ground-truth fluid volume are also reported for comparison. Moreover, we present the results of **future prediction (51-60 steps)**. Effectiveness and generalization ability of our method are proved.

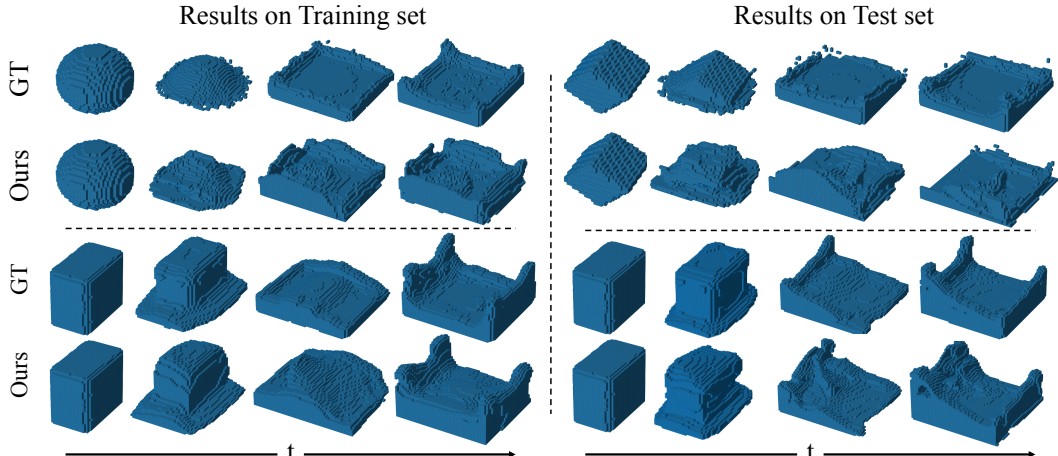

Figure 2: **Visualization results of predicted fluid volume**. Both results on training set and test set are shown. The results on training set prove that our method does capture geometry information of fluid from given images and performs the inverse rendering well. The results on test set fully demonstrate that our method learns **generalizable** fluid dynamics from videos.

truth is reported in Table 1, where the average error of all time steps is presented. The results on training set not only prove that our method infers accurate fluid geometry properties from images and performs **inverse rendering** well, but also demonstrate that our method learns fluid dynamics from videos. Meanwhile, our method shows strong ability to **generalize** to unseen data (test set) because our method learns real fluid dynamics explicitly from training videos, while most of related methods can only fit on one sequence every time. Then we compare our radiance-driven simulator with the simulator trained using ground-truth volume field. We can conclude from the results on test set that the simulator trained with videos as supervision performs comparably to the model trained with ground-truth volume field. We present some visualization results in Figure 2, where predicted fluid volume fields and ground-truth are shown. The volume field is visualized as voxel by setting a threshold. Accurate fluid volume fields are advected using our simulated velocity field. Following the common setting (Guan et al., 2022; Li et al., 2021b), we also compare our simulator with the model trained with GT in **future prediction** (51-60 steps). We roll out the simulator for 10 more steps that are not covered by the training set. Results are also reported in Table 1. Please refer to Appendix A for more results.

### 4.3 THE RESULTS OF RENDERING

We evaluate our renderer in this section. PSNR, SSIM (Wang et al., 2004) and LPIPS (Zhang et al., 2018) are used to evaluate the quality of the rendered images. We first report quantitative results of our method on test set in Table 2 and present rendered results in Figure 3. Our method achieves better results among most metrics and generalizes to test set without obvious deterioration even though the model has never seen these test images. Photorealistic images are synthesized using our method. To compare with other methods for rendering dynamic scenes and evaluate performance on novel view synthesis (NVS), we train two variants of NeRF (*i.e.*, T-NeRF and D-NeRF) on our

| Method | Fluid Fall (1 obj) | | | Fluid Fall (2 objs) | | | DPI Dam Break | | |
|---|---|---|---|---|---|---|---|---|---|
| | PSNR↑ | SSIM↑ | LPIPS↓ | PSNR↑ | SSIM↑ | LPIPS↓ | PSNR↑ | SSIM↑ | LPIPS↓ |
| | Results on training set | | | | | | | | |
| Ours | 37.45 | 0.981 | 0.163 | 36.11 | 0.973 | 0.188 | 38.54 | 0.986 | 0.141 |
| | Results on test set | | | | | | | | |
| Ours | 34.81 | 0.980 | 0.168 | 33.54 | 0.969 | 0.202 | 34.88 | 0.979 | 0.160 |
| | Results of NVS on test set | | | | | | | | |
| T-NeRF | 23.04 | 0.959 | 0.268 | 23.32 | 0.944 | 0.324 | 24.28 | 0.957 | 0.287 |
| D-NeRF | 25.94 | 0.963 | 0.305 | 25.50 | 0.949 | 0.333 | 25.05 | **0.960** | 0.307 |
| Ours | **32.87** | **0.976** | **0.194** | **31.81** | **0.967** | **0.221** | **32.03** | 0.959 | **0.197** |
| | Results of future prediction on test set | | | | | | | | |
| T-NeRF | 21.59 | 0.950 | 0.307 | 22.79 | 0.936 | 0.367 | 23.42 | 0.942 | 0.335 |
| D-NeRF | 24.01 | 0.959 | 0.295 | 25.38 | 0.944 | 0.349 | 24.49 | 0.946 | 0.328 |
| Ours | **33.72** | **0.980** | **0.178** | **29.73** | **0.958** | **0.282** | **29.59** | **0.959** | **0.215** |

Table 2: **Results of our method on rendering**. We present results of our method on training and test sets to prove the generalization ability of our method. Novel view synthesis and future prediction results on test set of our method, T-NeRF and D-NeRF are also shown. Note that T-NeRF and D-NeRF are trained on each sequence of test set, while our method is only trained on training set.

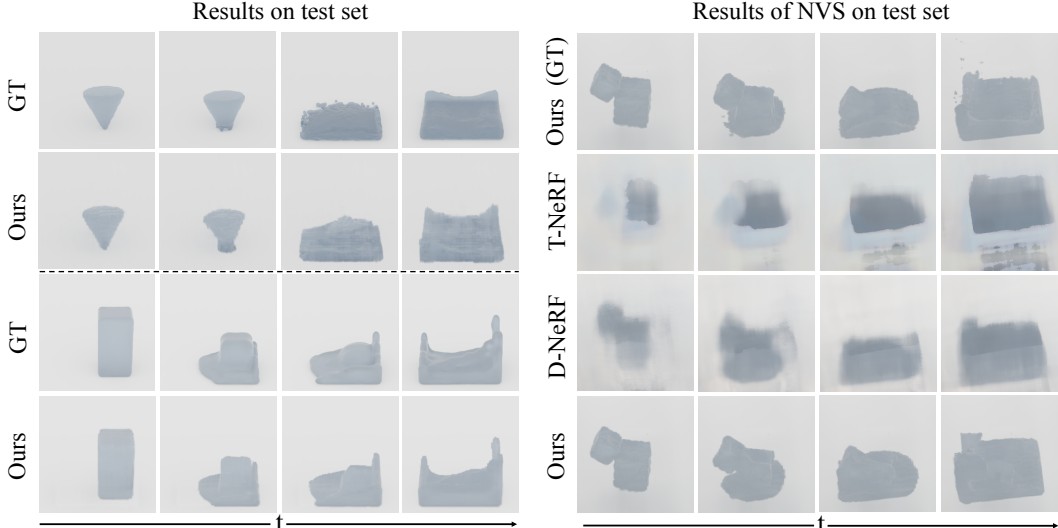

Figure 3: **Visualization of rendered images**. We present some results on test set to show generalization ability of our method and some examples of NVS on test set. Our method captures fluid dynamic and geometry properties of fluid and performs better on NVS than T-NeRF and D-NeRF. Ours (GT) denotes that we render the images with ground-truth volume field.

benchmarks. T-NeRF processes dynamic scenes by just combining the input position of sampled point $\mathbf{x}$ with an additional input time $t$. D-NeRF (Pumarola et al., 2021) takes an extra network to learn a transformation to canonical space for each sampled point at time step $t$, *i.e.*, $(\mathbf{x}, t) \longrightarrow \Delta\mathbf{x}$. Due to that D-NeRF and T-NeRF cannot generalize to unseen data, **we train D-NeRF and T-NeRF on each sequence of test sets respectively**. Our method is only trained on training set and evaluated on test set directly. As shown in Table 2 and Figure 3, our method achieves better performance and recovers fine details in both geometry and appearance. The generalization ability of our method is also fully proved. This indicates that our method reasons about real physical properties and geometric characteristics of the fluid. Future prediction experiments are also performed on test set using T-NeRF, D-NeRF and our method. The quantitative results are shown in Table 2.

## 4.4 COMPARISON WITH METHODS USING OTHER REPRESENTATIONS

We represent and simulate the fluid in an explicit and Eulerian way. NeuroFluid (Guan et al., 2022) is a concurrent work that represents fluid using particles. Such a way introduces/enhances two kinds

Generalize the trained model to unseen sequence

Figure 4: **Comparison with Lagrangian representation**. Results of experiments conducted under transfer setting are shown. Our method achieves significantly better performance and does learn generalizable fluid dynamics, while NeuroFluid collapses under such a transfer setting.

of ambiguity (*i.e.*, spatial ambiguity and temporal ambiguity) and thus formulates a prohibitively ill-posed inverse optimization problem. Detailed analyses of ambiguity are presented in Appendix A.5. In contrast, we represent fluid using the volume field and perform simulation in the Eulerian view. Thanks to this representation, accurate gradients with respect to fluid volume can be computed to update it directly and further drive the simulator with almost no ambiguity being introduced. Therefore, so many ambiguities make NeuroFluid can only overfit one sequence every time and cannot learn real fluid dynamics, while our model can be trained with many sequences together and learn generalizable fluid dynamics.

To demonstrate our statements, we train their model **using their published code** on the training set of our DPI dataset and report rendering results on the test set in Table 3. Our model generalizes to test set well and achieves significantly better performance. We also report the simulation results of NeuroFluid trained with videos and ground-truth particles respectively in Table 4 to prove that method NeuroFluid cannot infer dynamics from videos and transfer to unseen data. The average error of the particle positions with respect to ground-truth is used for evaluation. The visualization results of simulation and rendering are shown in Figure 4, where our results and NeuroFluid are both shown. We see that our model can be trained on multiple sequences simultaneously and transfers to unseen sequences, while NeuroFluid collapses when trained with multiple sequences and transferred to other sequences.

Moreover, we compare our method with (Li et al., 2021b) that represents fluid in the implicit space. This rough and implicit representation also limits its generalization ability as the authors of (Li et al., 2021b) also stated in their paper. We report rendering results on test set of our DPI dataset in Table 3. Our method achieves significantly better results.

| Methods | PSNR | SSIM | LPIPS |
|---|---|---|---|
| NeuroFluid | 27.89 | 0.948 | 0.244 |
| (Li et al., 2021b) | 24.23 | 0.933 | 0.317 |
| Ours | **34.88** | **0.979** | **0.160** |

Table 3: **Comparison with other methods**. Results of rendering are shown in the table.

| Methods | Average pos error (mm) |
|---|---|
| NeuroFluid | 1485.2 |
| CConv (train with GT) | 87.4 |

Table 4: **Simulation results of NeuroFluid**. The average position error of 50 steps is shown.

## 5 CONCLUSION

We propose an end-to-end and differentiable framework integrating fluid simulation, 3D modeling and rendering. The neural Euler simulator and NeRF-like renderer are associated using an explicit fluid volume representation. We achieve learning fluid dynamic model with videos as supervision by propagating error signals between rendered and ground-truth images to all modules. Various experiments involving simulation, NVS, future prediction and scene editing demonstrate both effectiveness and generalization ability of our method.

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

# A APPENDIX

## A.1 MODEL DETAILS

We first present detailed model architecture of our proposed neural Euler simulator (pressure projection solver). As shown in Figure 5, the inputs of the network contain three components: 1) the velocity field updated after advection and external force application; 2) the volume field of fluid used to provide space distribution of different materials (fluid and air); 3) distance field that each grid saves the normalized distance between its center and container boundary. These three parts are processed by three mini-networks respectively that contain two layers of 3D convolution (followed by BatchNormalization (Ioffe & Szegedy, 2015) and LeakyRelu (Maas et al., 2013)) and the output embeddings are then concatenated as a single feature. Finally, we send the concatenated feature into subsequent four convolutional layers and output $\Delta\mathbf{u}$ used to update velocity field.

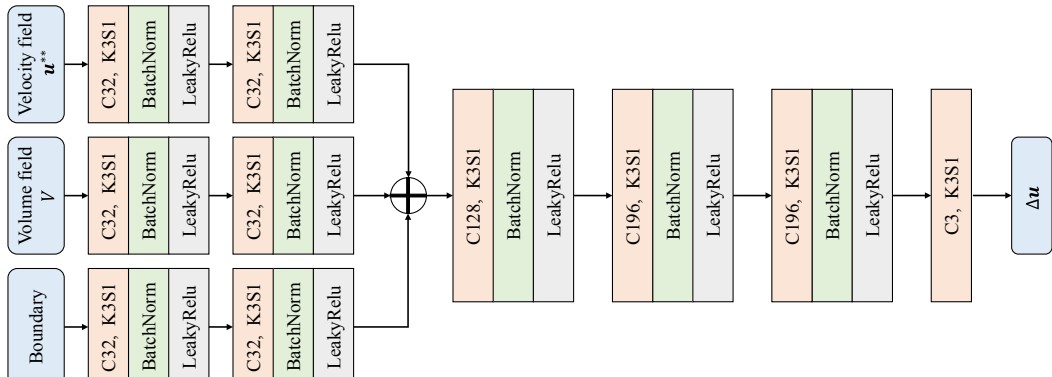

Figure 5: **Network architecture of our proposed pressure projection solver**. The network takes velocity field $\mathbf{u}^{**}$, the volume filed $V$ and boundary distance field as inputs. These three components are first processed by two layers respectively. Then the output features are concatenated and sent into subsequent layers for regressing $\Delta\mathbf{u}$. Each layer involves a 3D convolutional layer, a BatchNorm layer and a LeakyReLU (activation layer) except the last layer which only contains a convolutional layer. "C" denotes channel and "C32" represents that the number of channels is 32. "K" and "S" denote kernel size and stride respectively (*e.g.*, "K3S1" represents that the kernel size is 3x3x3 and the stride is 1).

Then we present detailed model architecture of our proposed volume-augmented neural radiance field (NeRF). As shown in Figure 6, the network is fully-connected and similar as NeRF (Mildenhall et al., 2020). In addition to point position $\mathbf{x} \in \mathbb{R}^3$ and corresponding camera view $\mathbf{d} \in \mathbb{R}^3$, we also send the fluid volume value $v$ of point $\mathbf{x}$ into the volume-augmented NeRF to regress density $\sigma$ and RGB color $\mathbf{c}$. Given the point $\mathbf{x}$, the fluid volume $v$ is generated by performing trilinear interpolation on fluid volume filed $V$ that is advected using simulated velocity field. The position vector $\mathbf{x}$ and view vector $\mathbf{d}$ are encoded using positional encoding before being sent into the NeRF.

## A.2 DATASET DETAILS

**Fluid Fall**. We generate two datasets by constructing a Fluid Fall environment. The simulation environment is where fluid objects fall in a cubic container. The size of the container is 2m x 2m x 2m. Each sample of the two version datasets contains one or two fluid objects randomly sampled from 5 different shapes (as shown in Figure 7). We simulate fluid fall data using a CFD tool Mantaflow (Pfaff & Thuerey). The viscosity is set to 0.004 for Fluid Fall (1 object) and 0.003 for Fluid Fall (2 objects). 60 time steps (50 steps for training) are simulated for each trajectory. The simulated fluid mesh is rendered using Blender. Two views for each sequence are generated. The ground-truth videos of these two datasets are rendered with different BRDF. The Blender project used for generating videos will be publicly available upon acceptance. To generate a dataset, we vary the initial shape, position and velocity. Each one of the two datasets contains 100 sequences. 90 sequences are used for training and the rest for test.

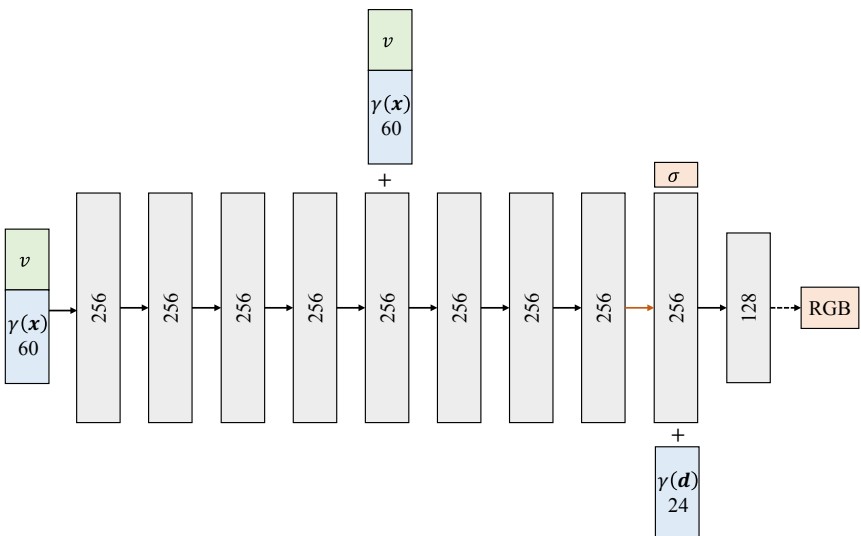

Figure 6: **Network architecture of our proposed volume-augmented NeRF**. The inputs contain positional encoded point $\mathbf{x}$, camera view $\mathbf{d}$ and fluid volume $v$. The outputs are density $\sigma$ and RGB color $\mathbf{c}$. All layers of the model are standard fully-connected layers, black arrows indicate layers with ReLU activations, orange arrows indicate layers with no activation, dashed black arrows indicate layers with sigmoid activation, and "+" denotes vector concatenation. We concatenate the positional encoded input location $\gamma(\mathbf{x})$ and the fluid volume $v$, and pass them through 8 fully-connected ReLU layers, each with 256 channels. We follow the NeRF (Mildenhall et al., 2020) architecture and include a skip connection that concatenates this input to the fifth layer's activation. An additional layer outputs the density $\sigma$ (which is rectified using a ReLU to ensure that the output density is nonnegative) and a 256-dimensional feature vector. This feature vector is concatenated with the positional encoding of the input viewing direction $\gamma(\mathbf{d})$, and is processed by an additional fully-connected ReLU layer with 128 channels. A final layer (with a sigmoid activation) outputs the emitted RGB radiance at position $\mathbf{x}$, as viewed by a ray with direction $\mathbf{d}$, given fluid volume value $v$ interpolated from volume field $V$.

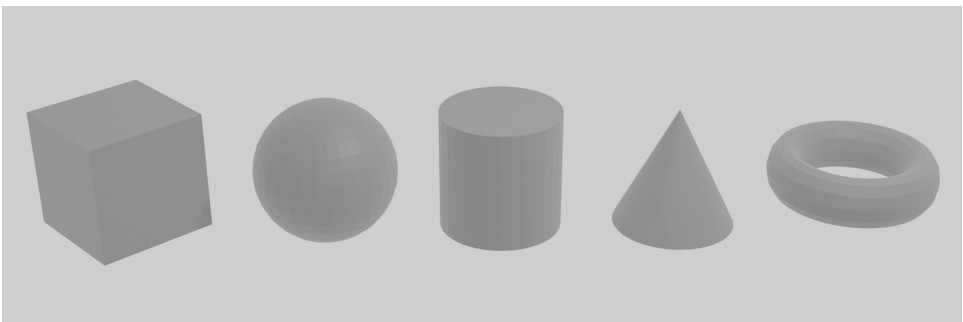

Figure 7: **The shapes of fluid objects**. We generate Fluid Fall (1 object) by randomly sampling one fluid body from the 5 objects shown in the figure and placing it into the container. The Fluid Fall (2 objects) is generated by randomly sampling two fluid bodies. Moreover, we vary the size, orientation, position and initial velocity of the fluid to generate datasets.

**Dam Break**. We randomly sample 180 sequences of 50 time steps for training and 20 sequences for test from the training set and test set of DPI Dam Break dataset (Li et al., 2019) respectively. This dataset is based on a Dam Break environment. The size of the container is 1.62m x 1.8m x 0.96m. The mesh of each time step is constructed from particles using Marching Cube algorithm and rendered using Blender. Two views that are the same as Fluid Fall datasets are generated for each sequence. The Blender project used for generating videos will be publicly available upon acceptance.

Advection in Lagrangian View    Semi-Lagrangian-based Advection in Eulerian View

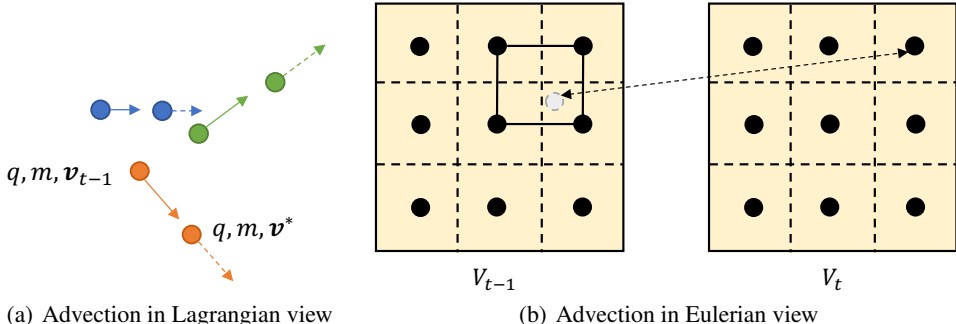

(a) Advection in Lagrangian view    (b) Advection in Eulerian view

Figure 8: **(a) Advection in Lagrangian view**. Particles move according to their velocities. The particles with dotted arrow are particles after applying advection. **(b) Advection in Eulerian view**. We perform advection to the volume field according the velocity field $\mathbf{u}_{t-1}$. The advection process contains two steps, *i.e.*, back-tracing and interpolation.

### A.3    ADVECTION MODULE DETAILS

We solve the complex dynamic equation by operator splitting. The time integration is split into three steps, *i.e.*, advection, applying external force and pressure projection. The advection can be seen as an operator that moves fluid with a constant velocity. The velocity field after applying advection and external force is not divergence-free, which violates the incompressibility of fluid. So the pressure projection is the key step to rectify the velocity and make the fluid looks real.

In the Lagrangian view, we achieve advection by just moving particles according to their velocity. The physical quantities such as mass and velocity are bound to particles and moved with these particles. As shown in Figure 8(a), there is a fluid object that contains three particles (*e.g.*, particle $q$) at time step $(t-1)$. We denote the mass and velocity of each particle as $m$ and velocity $v_{t-1}$ respectively. The velocity $v_{t-1}$ represents the average velocity over $(t-1) \rightarrow t$. The process of advection is just moving all the particles according to their velocities. The advected velocities are then applied with external force and pressure projection to get final velocities at the next time step $t$.

In the Eulerian view, the grids are fixed. We cannot apply advection as we do in the Lagrangian view directly. To achieve advection and update the physical quantities (volume and velocity) of each fixed grid, we take a commonly used semi-Lagrangian method to do advection. It contains two sub-steps, *i.e.*, back-tracing and interpolation. As shown in Figure 8(b), there is a volume field $V_{t-1}$. We now want to apply advection to the volume field $V_{t-1}$ and get the updated volume field $V_t$. The key is to compute the volume value of each grid center point in the field $V_t$. For each grid center point of field $V_t$, we first find where it is at time step $(t-1)$ and then borrow the volume value from $V_{t-1}$ to the grid center at time step $t$. The velocity field $\mathbf{u}_{t-1}$ represents the average velocity over $(t-1) \rightarrow t$. So we can back-trace from the grid center point at time-step $t$ using the velocity field $\mathbf{u}_{t-1}$ to find where it lies at time step $(t-1)$ and interpolate its volume value. Overall, the whole advection is achieved by back-tracing and interpolation, which are both differentiable. The velocity field is also advected in this way. But the advected velocities need to be applied with external force and pressure projection to get the final velocity field at the next time step.

### A.4    EXPERIMENTAL SETTINGS

The size of Euler grid is set to 40 x 40 x 40 for our generated datasets and 54 x 60 x 32 for DPI Dam Break. The image resolution is set to 256 x 256 and two views are used for training. The initial velocity field is generated by setting velocity of the grids to the initial velocity, and the initial volume field is generated by setting the volume value to 1.0 if the center point of grid is inside the initial fluid mesh otherwise to 0. We first pre-train the volume-augmented NeRF model using the first frames of all sequences for 20k iterations with a learning rate 5e-4 (decay 0.5 every 5k iterations). Then we jointly train the simulator and renderer with sequences data for 60k iterations. The initial learning

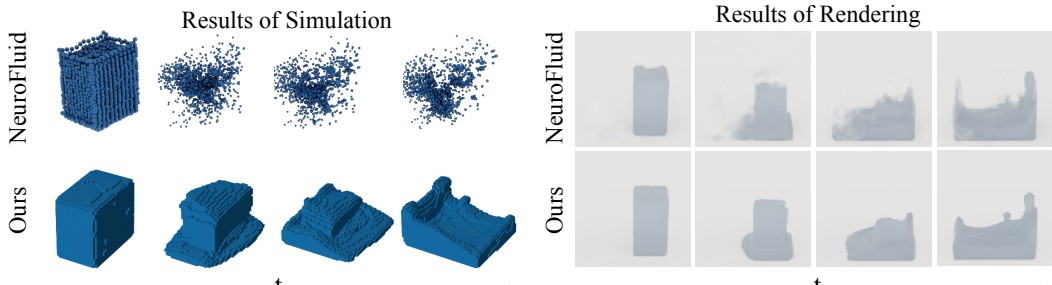

Figure 9: **Comparison with Lagrangian representation**. Results of experiments conducted under one sequence setting are shown. Our method achieves significantly better performance. Although NeuroFluid performs slightly better under one sequence setting compared to transfer setting, it is still hard for NeuroFluid to overfit on a more difficult sequence than what they use in their paper.

rate of the simulator is set to 5e-5 and the initial learning rate of the renderer is set to 1e-6. They are both decayed 0.5 every 10k iterations. The batch size is set to 4 for the joint training process. All models are trained on 4 NVIDIA RTX 3090 GPUs.

### A.5 AMBIGUITY ANALYSES OF DIFFERENT REPRESENTATIONS

As we have stated in the main paper, NeuroFluid (Guan et al., 2022) is a concurrent work that represents fluid using particles. Such a way introduces/enhances two kinds of ambiguity (*i.e.*, spatial ambiguity and temporal ambiguity) and thus formulates a prohibitively ill-posed inverse optimization problem. **Spatial ambiguity** exists between fluid particles and the image rendered using NeRF. NeRF constructs an implicit continuous field and renders image by sampling in this continuous space. However, particle representation is highly discrete and cannot be combined with NeRF directly. To make it continuous, NeuroFluid computes some complex and continuous distribution characteristics based on the discrete particles (through their proposed Particle-Driven NeRF (Guan et al., 2022)) to represent fluid. Then they can render image using NeRF to sample in this transferred and continuous field (*i.e.*, discrete particles → continuous field → image). Unfortunately, lots of spatial ambiguities between particles and the image are introduced by this transferring when doing inverse rendering, *e.g.*, many different combinations of particles may result in the same transferred field and further the same pixel color. Namely, even if we can infer the transferred fluid field from images, we cannot infer correct particle positions from the inferred field. Moreover, **temporal ambiguity** exists even if the inferred particles are accurate, *e.g.*, applying different motion vectors to the input particles may result in the same spatial distribution of particles at the next time step. Namely, no temporal correspondence is provided by the gradients from the image to help infer particle motion for Lagrangian simulation, which has to trace each particle in the whole life of the sequence. In contrast, we represent fluid using the volume field directly and perform simulation in the Eulerian view, which can be seen as a continuous field approximately through linear interpolation. Thanks to this representation, accurate gradients with respect to fluid volume can be computed to update it directly and further drive the simulator with almost no ambiguity being introduced. Therefore, so many ambiguities make NeuroFluid can only overfit on one sequence every time and cannot learn real fluid dynamics, while our model can be trained with many sequences together and learn generalizable fluid dynamics.

Note that all comparisons presented in the main paper are conducted under a transfer setting (*i.e.*, training on multiple sequences and testing on unseen sequences). For further comparison, we also train our model and NeuroFluid using only one sequence (under their setting). As shown in Figure 9, not a bad rendering result is achieved by NeuroFluid because their designed complex input features make its renderer to be overfitted more easily. However, such a modeling method cannot compute correct gradients for inverse optimization and learn fluid dynamics (simulation). A similar phenomenon is observed in Figure 3 and Figure 4 of their own paper (Guan et al., 2022). Note that it is harder for NeuroFluid to overfit on a more difficult sequence than what they use in their paper, even though we train it on only one sequence.

## A.6 ABLATION STUDY

Component analysis and parameter analysis are performed in this section. The effectiveness of our proposed radiance-driven fluid simulation is proved in Section 4.2. Our simulator shows comparable performance to the model trained with ground-truth Euler fluid grid. Then we drop the penalty used to regularize divergence of the updated velocity field during training and report the results on test set of Fluid Fall (1 obj) in Table 5 and Table 6. Such a regularization term stabilizes training and achieves significantly better simulation. We have mentioned in Section 3.4 that we first train the renderer and then jointly train the renderer and the simulator. We perform an ablation study to prove that the jointly training strategy does perform better than solely training the simulator and fixing the renderer. The results are also shown in Table 5 and Table 6. We finally analyze the improvement achieved by using more views for training. We train our model using 2 views and we compare it with the model trained with only one view. The results of the simulator and renderer on test set of Fluid Fall (1 obj) are shown in Table 5 and Table 6. Our method also works well when given videos with only one view, while more views bring obvious performance improvements in both simulation and rendering.

| Method | Fluid Fall (1 obj) | |
|---|---|---|
| | 1-50 steps | 51-60 steps |
| w/o Div. | 0.0548 | 0.0659 |
| w/o Joint training | 0.0398 | 0.0518 |
| One view | 0.0429 | 0.0521 |
| Ours | **0.0297** | **0.0362** |

Table 5: **Results of simulation**.

| Method | Fluid Fall (1 obj) | | |
|---|---|---|---|
| | PSNR | SSIM | LPIPS |
| w/o Div. | 26.16 | 0.963 | 0.276 |
| w/o Joint training | 31.03 | 0.966 | 0.172 |
| One view | 29.62 | 0.968 | 0.204 |
| Ours | **34.81** | **0.980** | **0.168** |

Table 6: **Results of rendering**.

We also analyze the robustness of our method to $\beta$ that is used to weight the contribution of divergence regularization term in equation 9. Experiments with different $\beta$ are conducted on Fluid Fall (1 obj) and the results of simulation are shown in Table 7. Our model is not very sensitive to different values of $\beta$. We further report more detailed results on Fluid Fall (1 obj) of training our model using different number of views in Table 8. In summary, more views result in better performance because more views provide more detailed 3D cues and more accurate and finer gradients for the simulator. However, our model also works well even with only one view.

| $\beta$ | 0 | 1e-6 | 1e-5 | 1e-2 |
|---|---|---|---|---|
| Grid Err. | 0.0548 | 0.0297 | 0.0322 | 0.0443 |

| #views | 1 | 2 | 5 | 10 |
|---|---|---|---|---|
| Grid Err. | 0.0429 | 0.0297 | 0.0234 | 0.0198 |

Table 7: **Analysis of $\beta$ used for optimization**.  Table 8: **Analysis of #views used for training**.

## A.7 SPACE AND TIME ANALYSIS

We report model size and timings in Table 9. The timing of inference is evaluated on one 3090 GPU. It takes about 48 hours to train the whole model on 180 sequences with four 3090 GPUs. Although improving the inference speed is not the main concern of this work, there are some tricks that could be taken to improve the rendering speed, *e.g.*, guiding the sampling according to the simulated fluid volume field.

| Models | Model Size (MB) | Model Size (#params) | Inference Time (per frame) |
|---|---|---|---|
| Simulator | 24 | 2,082,339 | 0.003s |
| Renderer | 15 | 1,307,464 | 2.6s |

Table 9: **Space and time analysis**. We present model size, the number of parameters and inference time of our simulator and renderer in the table.

## A.8 EXTREME VIEWS GENERATION AND SCENE EDITING

We explicitly simulate the fluid and the geometry of fluid is explicitly represented. Such a representation method endows the renderer with the ability to generate fluid videos from views that are far away from training distribution. To prove this, we train our model with **only one view** and synthetic images from **extreme views** (*e.g.*, overlook). Visualization results are shown in Figure 10(a).

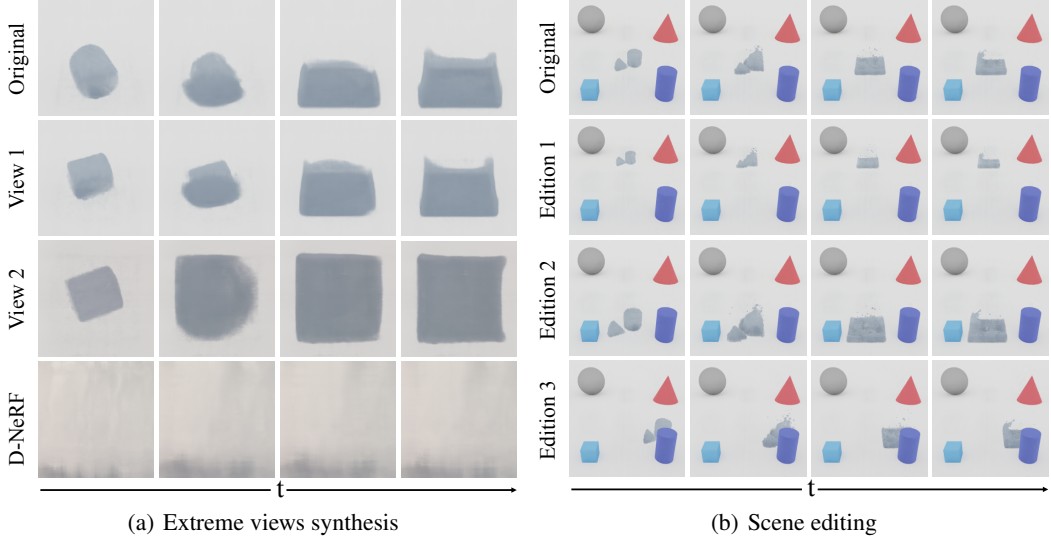

(a) Extreme views synthesis

(b) Scene editing

Figure 10: **(a) Images rendered from extreme views**. We train our model with only one view and render images from extreme views (*e.g.*, overlook). Our method still generates photorealistic images. **(b) Results of scene editing**. Our explicit fluid representation supports editing directly and handles occlusion well. We train the model on one sequence for simplicity on editing experiment.

Furthermore, one extra advantage of our fluid volume representation is that it is ideal and efficient for **scene editing**, *i.e.*, moving the fluid in the 3D space by simply applying a transformation to the volume field. As shown in Figure 10(b), we move the fluid in the scene and render corresponding images. We see that our model even handles occlusion well. Moreover, our simulated fluid can be easily placed into complex scenes by combining with other NeRF models.

## A.9   MORE VISUALIZATION RESULTS

We present more results of our simulated fluid volume field and rendered images on test sets of the three datasets in Figure 11. We develop an end-to-end framework integrating simulation, 3D modeling and rendering and learn fluid dynamics with videos as supervision. Although no ground-truth simulation data is provided, our method performs accurate simulation and renders photorealistic images. Rendered videos are shown in supplementary materials. Each video contains 60 frames, 10 frames of which are results of future prediction.

We also provide some results of novel view synthesis (NVS) on the test sets in Figure 12. Although we train our model on training set with only two views, our model generalizes to test set without obvious deterioration and synthesizes images from 50 different views that are out of training distribution. The camera rotates along z-axis and we render fluid of every step with different views. A part of views is shown in Figure 12. Rendered videos are shown in supplementary materials.

We have reported fluid editing (*i.e.*, moving the fluid in the scene) results in Figure 4(b) of our paper. We further rotate the fluid in the scene by directly rotating the simulated fluid volume field during inference. Visualization results are shown in Figure 13, where the fluid falls, translates and rotates in the scene. Videos of scene editing are shown in supplementary materials. Note that the model used to perform scene editing is trained on one sequence with two views.

## A.10   LIMITATIONS AND FUTURE WORK

Same as most of other methods for inverse rendering (Guan et al., 2022; Munkberg et al., 2022; Hasselgren et al., 2022; Yu et al., 2022), we mainly focus on learning from synthetic data. It remains a lot of great challenges from many research areas (*e.g.*, differentiable photorealistic rendering, optics, *etc.*)  and requires extensive research efforts as well as technical development to achieve learning from real-world data, especially for fluid that contains complex light-transport effects (*e.g.*,

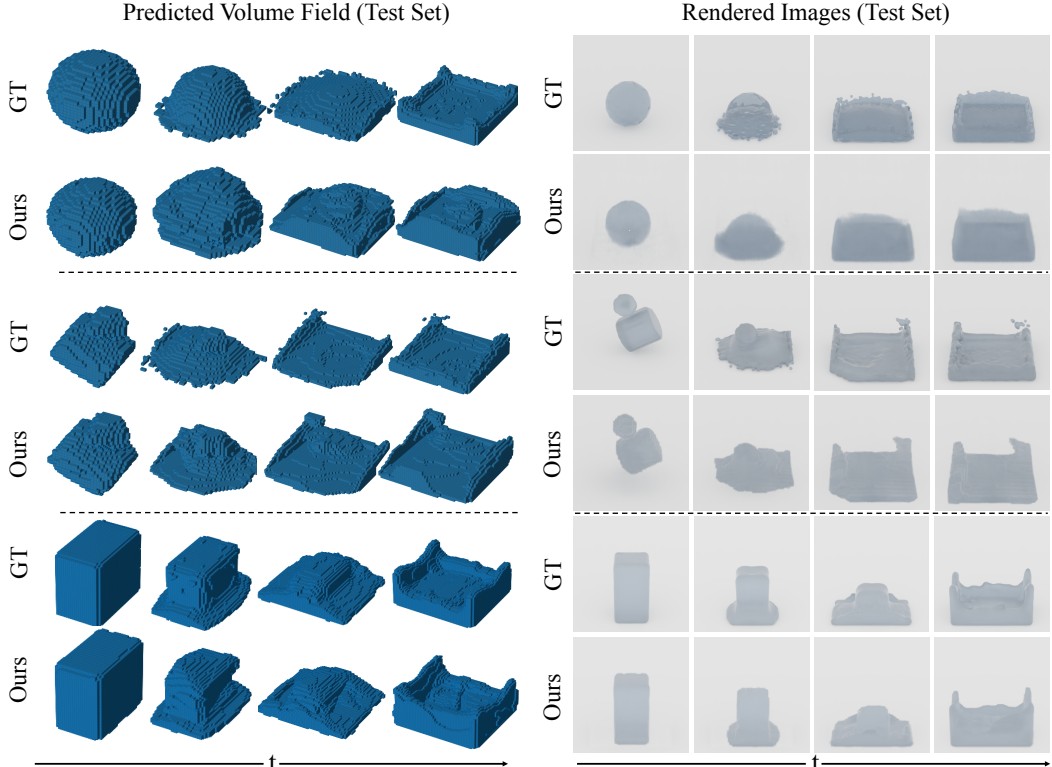

Figure 11: **Some predicted volume field and rendered images on test set of the three datasets**. We train our model on training set and present the results on test set to prove effectiveness and generalization ability of our method. Lines 1-2 present results on Fluid Fall (1obj). Lines 3-4 show results on Fluid Fall (2objs). Lines 5-6 present results on DPI Dam Break.

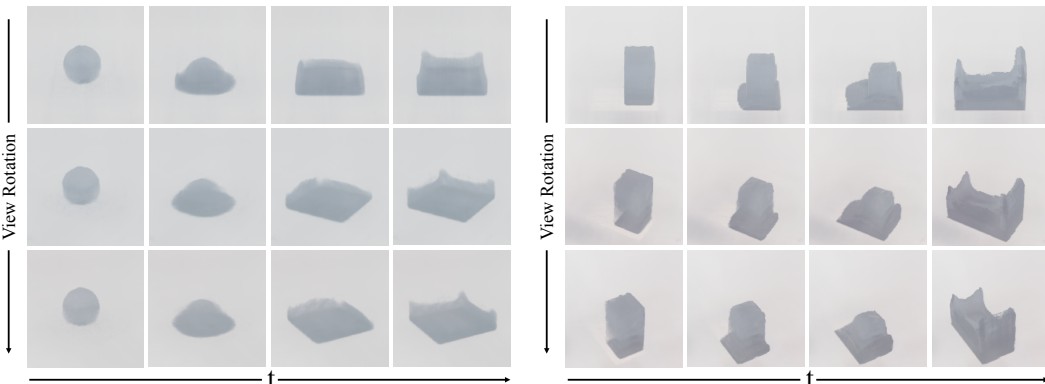

Figure 12: **Some visualization results of NVS on test sets**. Although we train our model on training set with only two views, the model generalizes to test set well and our explicitly represented fluid facilitates the renderer to synthesize images from views that are out of training distribution. Videos are provided in supplementary materials.

refraction, reflection, *etc.*). Although some differentiable renders (Wang et al., 2022; Bemana et al., 2022) are proposed to consider refraction in fluid and perform **forward** rendering well, lots of ambiguities are introduced by refracting (*e.g.*, different local shapes may result in the same local color) and make the optimization of **inverse** rendering task exceedingly difficult. Thereby, it still remains a great challenge to do inverse rendering for fluid data from real world and beyond the current state-of-the-arts. A potential solution is to extract and utilize geometry cues contained in

Fluid Editing (translation and rotation)

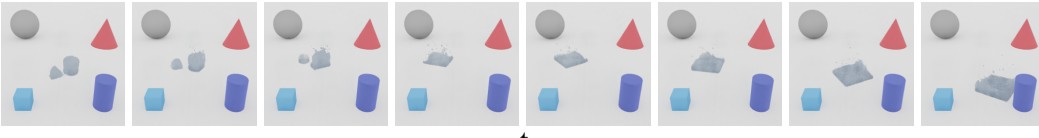

Figure 13: **Results of scene editing**. In addition to translation, we apply rotation to the fluid by directly rotating the simulated fluid volume field during inference. Please refer to videos provided in supplementary materials for detailed results.

images to reduce ambiguities and infer fluid dynamics. Besides, embedding fluid-solid coupled simulation in our framework and using our model to infer external forces from videos are also interesting research topics.

