# OpenReview forum: "Inferring Fluid Dynamics via Inverse Rendering"
_ICLR.cc/2023/Conference — Submitted to ICLR 2023_

### Official Review · Reviewer_DkWW · 2022-10-19

**Confidence:** 5
**Correctness:** 4
**Technical Novelty And Significance:** 4
**Empirical Novelty And Significance:** 4
**Recommendation:** 8

**Clarity, Quality, Novelty And Reproducibility:**

While there exist some other papers that propose solutions to similar problems, the solution proposed by this paper is quite novel. However, this paper has some major clarity issues.

First of all, there are significant grammatical errors in the paper. For example, section 4.1 sentence 5 reads “We generate two versions datasets…” Section 4.2 sentence 5 reads in part “...to unseen data … due to that our method does learn fluid dynamics…” A thorough proofreading pass will be necessary before publication.

Second, it would be difficult to reproduce this paper because the methodology isn’t explained in enough detail. For example, the advection process only cites another paper, but doesn’t say how this paper implements it. Furthermore, section 3.3 says that the volume field is advected based on the velocity field, but doesn’t describe how that is done. Given that section 3.1 paragraph 3 makes it sound as if the use of a volume field is unique to this paper, knowing how it is updated is critical to reproducibility.

Finally, the figures in the paper are difficult to understand. For example, Figure 1 doesn’t make sense. There are 2 advection boxes on the volume field path and 1 on the velocity field path. Are they doing the same thing? Presumably not, but that’s not clear. Also, the velocity field is an input to the advection box between V_t and V_{t+1}, but not an input to the advection box between V_{t-1} and V_t? Also, the gradient flow seems to show the gradients going straight from the NeRF to the pressure ConvNet when they actually pass back through the advection step. Furthermore, Figures 2, 3 (left), and 9 have alternating rows GT and Ours. Presumably each pair of rows are just more examples, but the way they’re presented makes the reader think they are different methods being compared. It would make the figures more clear if the GT-Ours comparison rows could somehow be grouped, e.g., put a line between rows that aren’t the same scene.


**Strength And Weaknesses:**

The biggest strength of this paper is that the proposed methodology is able to learn fluid dynamics from videos alone! Recent work has shown how this can be done for static scenes and even small amounts of movement, but fluids are high dimensional and have quite complicated dynamics. It is remarkable that this is able to be done, and done well according to the results, from only video data. What this paper is missing is a video attachment, which I suspect would get the results across much better than the static images. An additional strength of this paper is both the analysis of the pressure ConvNet in isolation of the NeRF (by using the GT fluid state, which is available because this is all in sim), and the comparison to T-NeRF and D-NeRF.

The paper does have a few areas for improvement, however. First of all, the separation of responsibilities between the advection method used and the learned pressure projection is unclear. This makes it hard to assess what the network is learning. Is it a hard task? Or is the advection algorithm doing most of the heavy lifting? This can be fixed by 1) putting a summary of the advection algorithm in the methodology section, and 2) putting a detailed mathematical description of the algorithm in the appendix. Second, the apparent train/test data is fairly simplistic. Most of it appears to be fluid in a cube enclosure, with the main variation being the starting configuration. Note the “apparent.” The figures in the paper show only these settings. If there are more complex settings, it is not obvious. This can be fixed by either featuring already existing complex settings in figures, or by adding such settings to the datasets. Finally, the biggest baselines to the proposed methodology are buried in the appendix (A.4). These are the main alternatives to what the paper proposes, their comparisons should be featured prominently in the results section (if space is an issue, the details of the comparison can remain in the appendix).

Also, one additional comment: The data used in this paper is all sim data. Given that the methodology only requires video, why not try it on a real world dataset? I think those results would be very interesting.


**Summary Of The Paper:**

This paper approaches the problem of learning fluid dynamics with a neural network. To do this, the paper proposes combining a NeRF with the dynamics network to do joint training. The fluid is represented on a grid (the Eulerian way) as volume and velocity. First, the fluid is advected and gravity applied via the standard fluid equations. Next, the fluid is fed into a 3D ConvNet to compute the pressure projection. Finally, the scene is rendered with a NeRF, which takes the volume of fluid at a given location as an additional input. The results show that the joint system is able to accurately model fluid dynamics and render realistic images of the fluid.

**Summary Of The Review:**

Despite my above concerns, this is actually a very good paper. It solves a difficult task from limited data, and does so well. The paper could use some work on clarity and details, but overall the proposed methodology is novel and the evaluation is thorough.

---

> ### Author Response · Authors · 2022-11-09
> **Response to Reviewer DkWW**
>
> We sincerely thank you for your professional advice and appreciation of our work. We have revised our paper following your suggestions and submitted the **rebuttal revision**. We also resubmit our rendered videos as supplementary materials.
>
> **For Weakness 1:** We add a brief explanation of the advection in Section 3.1 of our main paper and a detailed description of this module in Appendix A.3.
>
> **For Weakness 2:** We do use only cube enclosure to generate fluid data. But in theory, our approach could easily be applied to more complex settings. We will add more complex settings to the datasets as suggested and make them publicly available.
>
> **For Weakness 3:** Following your suggestions, we put the main results reported in Appendix A.4 in the main paper (Section 4.4) and keep the detailed discussions in A.5. Then we move the experiments of extreme view generation, scene editing and ablation studies to Appendix.
>
> **For the additional comment:** It is an insightful opinion to directly apply our method to real-world datasets. As for well-controlled lab scenario, we think our method can work without modification. However, currently no such datasets are publicly available. Collecting our own real-world dataset requires a lot of work and high-cost components like high-speed cameras and suitable lab space. But it is an interesting topic and we would like to focus on it in the future. As for scenarios in the wild, it remains lots of great challenges to perform reverse rendering using data in the wild (especially fluid data) as we discussed in Appendix A.10 (rebuttal revision).
>
> **For clarity issue 1:** We have fixed the grammatical errors you mentioned, and we are keeping to proofread our paper as suggested.
>
> **For clarity issue 2:** As shown in the rebuttal revision, we have added a detailed explanation of the advection to make this part clearer.
>
> **For clarity issue 3.1:** The advection can be seen as an operator that moves fluid with a constant velocity. In the Lagrangian view, we achieve advection by just moving particles according to their velocity. The physical quantities such as mass and velocity are bound to particles and moved with these particles. However, the grids are fixed in the Eulerian view. To achieve advection and update the physical quantities (volume and velocity) of each fixed grid, we take a commonly used semi-Lagrangian method (back-tracing and interpolation) to do advection as we have explained in the rebuttal revision. As shown in Figure 1 and equations 3-6 of our paper, we first use the velocity u_{t-1} to do advection and get V_{t} and u*. By further applying external force and pressure projection, we get the velocity field in the next time step u_{t}. We now have to associate the renderer with the updated velocity u_{t} and train the pressure ConvNet. To achieve this, we continue to update the next time step, i.e., update V_{t} to V_{t+1} by using the output velocity u_{t} to advect the volume V_{t}. The advected volume V_{t+1} is sent into the renderer to generate an image and the gradients go from the NeRF to the advection module and pressure ConvNet finally. Note that the advection module contains no learnable parameters and is fully differentiable. We apologize for the lack of clarity on this part, but the time integration we took is a commonly used formulation and we are sure it is right. Please let us know if you have some new suggestions to help us make it clearer.
>
> **For clarity issue 3.2:** As shown in the rebuttal revision, we have polished the figure.
>
> Thanks again for your golden suggestions. Please let us know if you still have any unclear parts or advice about our work. We would be absolutely delighted to further discuss with you and improve our paper.

---

### Official Review · Reviewer_24gb · 2022-10-25

**Confidence:** 4
**Correctness:** 2
**Technical Novelty And Significance:** 3
**Empirical Novelty And Significance:** 3
**Recommendation:** 5

**Clarity, Quality, Novelty And Reproducibility:**

Clarity: Fair. My main concern is technical clarity. See weakness above.

Quality: Overall the quality is good.

Novelty: I believe incorporating simulator with differentiable rendering is novel.

Reproducibility: Fair.

**Strength And Weaknesses:**

+ It is in general an interesting idea to incorporate explicit simulator with differentiable rendering to allow inference from videos.
+ Approximating pressure solve with a CNN looks promising.
+ It is interesting to see plausible future prediction results from novel views.

- My main concern is on technical clarity. It is unclear how the generalization and test-time inference is done. In evaluation, it is emphasized that the proposed method is trained on training set only and directly used on test set, while other methods need test-time optimization. But I don't see an inference module in the architecture. It seems like the approach needs a conditional NeRF and two volumes for velocity/occupancy. How are these components inferred given a test-time video without any optimization?
- How is the joint training done? Does it train on separate frames, or train on the last frame only?
- How is the pre-training for NeRF done? It seems that the NeRF needs volume as input, and if it is trained on first several frames, then how can it be trained "without any groundtruth fluid dynamics"?
- How to generate the velocity field for training and testing? Since it is claimed that the model is trained without dynamics, I assume it is randomly initialized at the very beginning of training/testing each scene instance, and it gets evolved across time?
- Since the proposed model already has an explicit representation of the fluid, why does it need a NeRF for rendering instead of directly rendering the volume? Is resolution the main concern?
- It is slightly confusing that in Figure 3 the GT frames have a different color for BG/FG compared to the results (also this happens for later figures), while the PSNR in Table 2 looks high. Is it an issue with visualization in figures?
- Please use \citep{} for references.

**Summary Of The Paper:**

This paper presents an approach to inferring 3D fluid dynamics from 2D videos. The main idea is to embed a fluid simulator and explicitly represent fluid as a volume, augmented by a NeRF-like volume rendering framework to bridge the 3D representation to 2D videos. Experiments show good results on synthetic liquid datasets.

**Summary Of The Review:**

I think the paper presents an interesting attempt to incorporate differentiable simulator with differentiable rendering for inferring fluid dynamics. However, my main concern is the technical unclarity that makes it difficult to evaluate the overall quality. I'm willing to raise my score if the technical clarity can be substantially improved.

---

> ### Author Response · Authors · 2022-11-09
> **Response to Reviewer 24gb [Part 2/2]**
>
> **For Weakness 1:** The training and inference process of our method share the same architecture. The inference is performed under a user-defined scenario, i.e., the user can specify initial conditions that are totally different from the training set and perform simulation and rendering using our trained model. We didn't say that other methods need test-time optimization. We said that other methods train the model using only one video every time and would collapse when given initial conditions different from the training video during inference. Our method can be trained on many videos simultaneously and learn explicit fluid dynamics. This makes our simulator work well when given unseen initial conditions. The renderer also constructs an accurate mapping between the simulated volume field and images during training. So the whole system can work well when given initial conditions that are different from the training set without test-time optimization.
>
> **For Weakness 2:** We jointly train the simulator and renderer on consecutive frames. In other words, the model is trained in a roll-out fashion. For each sequence, we send the initial volume and velocity (i.e., $V_0$ and $\mathbf{u}_0$) field into the simulator and predict $V_1$ and $\mathbf{u}_1$. Then the predicted $V_1$ and $\mathbf{u}_1$ are sent into the simulator to predict $V_2, \mathbf{u}_2$ and so on. In the meantime, each predicted volume field is also taken as the input of the renderer to predict image that is supervised by the given video.
>
> **For Weakness 3:** During pretraining, we do need the volume of the first frame, but **not the first several frames**. The volume of the first frame is necessary for starting the simulation (all simulators need initial condition to start simulation) but it does not provide any dynamic information. So we said “without any ground-truth fluid dynamics”. If you have any suggestions on the wording of this part, we would be delighted to discuss with you.
>
> **For Weakness 4:** In our setting, we provide the velocity field of the first frame for starting the simulator during training. In the scenario that we cannot get initial conditions directly, we give a potential solution in above explanation part.
>
> **For Weakness 5:** The primary function of the NeRF model is to provide accurate gradients from videos and support inverse rendering when we train our system. During inference, the NeRF model serves as a renderer that naturally supports end-to-end simulation and rendering. Of course, most renderers (whether differentiable or not, whether learning-based or classical) can be used to render the simulation results during inference.
>
> **For Weakness 6:** As we see from Figure 3, our model generates similar BG colors and slightly different fluid colors compared to GT. Overall, the whole generated image (layout and shape of the fluid) is similar to GT. The quantitative results corresponding to this part were reported in Table 2. In extreme view generation (as shown in Figure 4 of the original version or Figure 10 of the rebuttal revision), the model does generate different colors compared to GT because we train our model on only two views that are quite different from the views used for inference. We did not report the quantitative results of extreme view generation in Table 2, where only the results of normal/training views and novel views were reported.
>
> **For Weakness 7:** Thanks very much for your suggestion. We have modified the references as shown in the rebuttal revision.
>
> Thanks again for your professional comments. Please let us know if you still have any unclear parts of our work. We would be absolutely delighted to further discuss with you.

---

> ### Author Response · Authors · 2022-11-09
> **Response to Reviewer 24gb [Part 1/2]**
>
> We thank you for your golden comments, which help us improve the quality of this paper.
>
> Before addressing your concerns, we first make a detailed explanation of our method.
>
> We generate fluid videos by using a physical engine with different initial conditions (shape, velocity and position) and render the simulated results as videos. These videos and corresponding initial conditions are provided to train the model. The initial volume and velocity (only the first frame) are used to start simulation during both training and inference. During training, we reuse the initial volume to pretrain the renderer first and then jointly train the simulator and the renderer given the initial conditions and videos. The initial conditions are used to start simulation and the videos are used as supervision during joint training. During inference, it is a user-defined scenario, i.e., users set arbitrary initial shape and velocity (which can be different from the training set), then our system outputs predicted fluid dynamics and videos.
>
> Note that all physical simulators, whether classical or learning-based, need initial conditions (e.g., initial position, volume, velocity, etc.) to start simulation/prediction. We just reuse the initial volume field to pretrain the renderer. Thereby, the limitation of needing an initial volume field does not (only) come from our training strategy.
> In our setting, we provide the volume of the first frame of all training sequences. In the real-world lab scenario, we can also set the initial shape and velocity of the fluid to generate training data and record the initial conditions. In the scenario that we cannot get initial conditions directly, there are two potential solutions: 1) randomly initialize the initial conditions and optimize them during training as you suggested. 2) reconstruct the initial conditions from the given videos. However, as proved in many static inverse rendering works, random initialization makes the whole system completely without prior information injection and the model is almost impossible to converge. And reconstructing very accurate initial conditions is also a hard task. Naturally, a feasible and commonly used solution is to combine these two solutions, i.e., reconstruct coarse initial conditions and optimize them through training.

---

> ### Author Response · Authors · 2022-11-16
> **Look forward to your feedback**
>
> We sincerely thank you for the review and comments. We have provided corresponding responses and explanations, which we believe have covered your concerns. We hope to further discuss with you whether or not your concerns have been addressed. Please let us know if you still have any unclear parts of our work.

---

### Official Review · Reviewer_zis7 · 2022-10-26

**Confidence:** 4
**Correctness:** 4
**Technical Novelty And Significance:** 2
**Empirical Novelty And Significance:** 2
**Recommendation:** 5

**Clarity, Quality, Novelty And Reproducibility:**

As far as I see, the paper provides sufficient details to reproduce the experiments. The paper’s novelty is a little bit limited because of previous works that combined differentiable physics and differentiable rendering.

**Strength And Weaknesses:**

Strengths

- I appreciate the differentiable physics and the differentiable renderer, where the simulator is further augmented with a neural network. It provides the framework with the ability to generalize across different scenes. This approach is effective and will inspire future work.
- Learning fluid dynamics will be of practical relevance.
- I like the explanation of the Lagrangian v.s. Eulerian representation. The baseline comparison is convincing. Eulerian representation is easier for implementing a fluid simulator and is very natural to combine with NERF.

Weakness

- It seems that the method requires an initial volume field for pretraining the NERF so that it can start the simulator. Will this limit the application in the real world? After all, obtaining ground truth volume in the real world is very hard. Can the volume be estimated through NERF directly?
- Though Eulerian representation simplifies the fluid simulator, it's a costly representation compared with particles (for training), and one has to predefine the volume's resolution and size. What if we combine the Langrangian and Eulerian views as in MPM?
- The lack of real-world experiments kind of weakens the experiment. The authors have explained this limitation in the appendix, which will largely reduce the paper's contribution.

**Summary Of The Paper:**

This paper proposes a way of combining NERF-based volumetric renderer and eulerian fluid simulation to reconstruct fluid dynamics from videos. Given synthesized fluid videos, it can use NERF to estimate the volume of fluid from the images and then simulate it with a differentiable Euler simulator with a ConvNet as the projection solver. The volume in the simulator is fed as an additional input to the volumetric renderer, enabling supervising dynamics with videos directly. Authors conduct experiments on synthetic data and demonstrate effectiveness against NERF-based approaches without physical priors.

**Summary Of The Review:**

This paper provides a simple yet effective approach to combine NERF-based differentiable rendering and an Eulerian fluid simulation. However, the methods are not very applicable in the real world, and the novelty is limited. From my perspective, the paper is marginally below the acceptance threshold.

---

> ### Author Response · Authors · 2022-11-09
> **Response to Reviewer zis7**
>
> Thanks very much for your invaluable comments and appreciation of our work.
>
> **For Weakness 1:** All physical simulators (classical and learning-based) need initial conditions (e.g., initial volume, velocity, etc.) to start simulation. We just reuse the initial volume field to pretrain the NeRF and endow it with the ability to do inverse rendering preliminarily. Thereby, the limitation of needing an initial volume field does not (only) come from our training strategy. In the real-world scenario like a well-controlled lab scenario, we can manually set the initial volume or we can reconstruct the initial volume using NeRF with images from multiple views as input. For example, we can take all grid centers of the volume field as input of NeRF to estimate the density values and then reconstruct the initial volume field. However, it remains great challenges from many research areas and beyond the current state-of-the-arts to reconstruct fluid volume in the wild scenario as we have discussed in Section A.8 of our paper. But there is a potential solution that we can reconstruct coarse initial volume from multi-view images and optimize them by training.
>
> **For Weakness 2:** It’s a great idea to combine Eulerian and Lagrangian representations. Actually, we have tried to borrow the design philosophy of flip and designed a leaning-based/differentiable simulator using hybrid representation. It works well on forward simulation, however, such a hybrid representation also introduces large amounts of temporal and spatial ambiguities and makes the inverse problem hard to converge. Hence we have to consider both the performance of forward simulation and the ability of reverse rendering when we design the simulator. In general, we think that representing fluid in the Eulerian view to do the inverse task is the best choice at present. But it still remains an interesting topic to design hybrid and stronger simulators that support inverse rendering well.
>
> **For Weakness 3:** We totally agree with your insightful opinions on extending our method to real-world scenarios. However, to make it happen, a lot of challenges coming from other research areas (such as differentiable photorealistic rendering, optics, etc.) need to be solved, which themselves require extensive research efforts as well as technical development. With the current status of research, it is nearly impossible to learn fluid dynamics directly from data in the wild. Therefore, currently ALL existing methods [1, 2, 3, 4, 5, 6, 7] only learn fluid dynamics from synthetic data.
>
> Despite these obstacles, our proposed method makes an important progress in this community which learns fluid dynamic model directly from videos. Although there are some related works that also combine differentiable physics and differentiable rendering, as discussed in our paper, most of them just memorize training sequence(s) through training. Our work is the first to develop a system that supports learning **generalizable** fluid dynamics model from videos. We think this is a big step to bridge the gap between learning 3D dynamic model and accessible 2D videos, and opens up possibilities for learning from real-world data. We therefore think it is unfair for us that judging the value of the work based on whether or not it could be directly applied to real-world scenarios currently.
>
> **For the concern about novelty**: Although there are some works [1, 2, 8] that combine differentiable physics and differentiable rendering, as we discussed in our paper, they either only consider rigid body dynamics or overfit the training sequence(s). In our work, we give detailed **ambiguity analyses** of different representations, which are crucially important for designing a inverse system but are overlooked by previous work. Based on this, we are the first to design an end-to-end system that supports learning **generalizable** and **explicit** fluid dynamics model from videos.
>
> [1] Guan S, Deng H, Wang Y, et al. NeuroFluid: Fluid Dynamics Grounding with Particle-Driven Neural Radiance Fields. In ICML 2022.
>
> [2] Li Y, Li S, Sitzmann V, et al. 3d neural scene representations for visuomotor control. In CoRL 2022.
>
> [3] Sanchez-Gonzalez A, Godwin J, Pfaff T, et al. Learning to simulate complex physics with graph networks. In ICML 2020.
>
> [4] Li Y, Wu J, Tedrake R, et al. Learning Particle Dynamics for Manipulating Rigid Bodies, Deformable Objects, and Fluids. In ICLR 2019.
>
> [5] Ummenhofer B, Prantl L, Thuerey N, et al. Lagrangian fluid simulation with continuous convolutions. In ICLR 2020.
>
> [6] Pfaff T, Fortunato M, Sanchez-Gonzalez A, et al. Learning mesh-based simulation with graph networks. In ICLR 2021.
>
> [7] Tompson J, Schlachter K, Sprechmann P, et al. Accelerating eulerian fluid simulation with convolutional networks. In ICML 2017.
>
> [8] Jiajun Wu, Erika Lu, Pushmeet Kohli, Bill Freeman, and Josh Tenenbaum. Learning to see physics via visual de-animation. In NeurIPS 2017.

---

### Decision · Program_Chairs · 2023-01-20

**Decision:**

Reject

**Justification For Why Not Higher Score:**

Two reviewers raised the

**Justification For Why Not Lower Score:**

N/A

**Metareview: Summary, Strengths And Weaknesses:**

This paper proposes to infer the dynamics of fluid by integrating differentiable simulation and rendering. Reviewers all agree that the problem is worth studying, but have concerns regarding the presentation and the limited experiments. After the rebuttal, there was extensive discussion about the paper among all reviewers and the AC. Two reviewers are negative, and one is positive. The main point of disagreement is whether the assumption of the initial state is realistic. The negative reviewers argued that if such a perception module exists, the value of learning the dynamics model becomes moot. At the same time, the positive reviewer suggests that the entire framework may be considered in a Bayesian filtering framework.

The AC had discussions with the reviewers and agreed on the concern. To be considered a Bayesian filtering framework, this submission should demonstrate how the learned dynamics model can be integrated with updates from the perception model, which is currently missing. If the main argument of the paper is to learn the dynamics model directly from video via neural rendering, then the assumption of the initial state greatly undermines the value of the paper, as the introduction of differentiable neural rendering is supposed to address the limitation. Therefore, the AC agrees with the negative reviewers that this is a significant concern, making the paper below the bar to be accepted.  The authors are encouraged to revise the paper based on the reviews for the next venue.